# Physiological and Clinical Responses in Pigs in Relation to Plasma Concentrations during Anesthesia with Dexmedetomidine, Tiletamine, Zolazepam, and Butorphanol

**DOI:** 10.3390/ani11061482

**Published:** 2021-05-21

**Authors:** Anneli Rydén, Marianne Jensen-Waern, Görel Nyman, Lena Olsén

**Affiliations:** Department of Clinical Sciences, Swedish University of Agricultural Sciences, SE-750 07 Uppsala, Sweden; marianne.jensen-waern@slu.se (M.J.-W.); gorel.nyman@slu.se (G.N.); lena.olsen@slu.se (L.O.)

**Keywords:** animal welfare, pharmacokinetic, swine, transportation, short term

## Abstract

**Simple Summary:**

Reliable protocols are needed for short-term anesthesia in pigs. The study’s aim is to identify an anesthetic procedure that, without the use of sophisticated equipment, ensures an acceptable depth and length of anesthesia, a regular spontaneous breathing pattern, and a stable hemodynamic condition for the animal. A total of 12 pigs were given a single intramuscular injection of dexmedetomidine, tiletamine, zolazepam, and butorphanol. To investigate the possibility of prolonging the anesthesia, six of the pigs also received an intravenous dose of the drug combination after one hour. Physiological and clinical responses and drug plasma concentrations were examined. The main results suggest that intramuscular administration of the drug combination provides up to two hours of anesthesia with stable physiological parameters and an acceptable level of analgesia. An intravenous administration of one-third of the original dosage prolonged the anesthesia for another 30 min. Since the pigs were able to breathe spontaneously, none of them were intubated. The study also provides new information about each drug’s plasma concentrations and the impact of the drug combination in pigs. This technique can be used to perform nonsurgical operations or transports when short-term anesthesia is required.

**Abstract:**

Reliable protocols for short-term anesthetics are essential to safeguard animal welfare during medical investigations. The aim of the study was to assess the adequacy and reliability of an anesthetic protocol and to evaluate physiological and clinical responses, in relation to the drug plasma concentrations, for pigs undergoing short-term anesthesia. A second aim was to see whether an intravenous dosage could prolong the anesthesia. The anesthesia was induced by an intramuscular injection of dexmedetomidine, tiletamine-zolazepam, and butorphanol in 12 pigs. In six of the pigs, a repeated injection intravenously of one-third of the initial dose was given after one hour. The physiological and clinical effects from induction to recovery were examined. Plasma concentrations of the drugs were analyzed and pharmacokinetic parameters were calculated. Each drug’s absorption and time to maximal concentration were rapid. All pigs were able to maintain spontaneous respiration. The route of administration did not alter the half-life of the drug. The results suggest that intramuscular administration of the four-drug combination provides up to two hours of anesthesia with stable physiological parameters and an acceptable level of analgesia while maintaining spontaneous respiration. A repeated intravenous injection may be used to extend the time of anesthesia by 30 min.

## 1. Introduction

Pigs have genetic, anatomical, and physiological similarities to humans, which makes them one of the most useful and versatile animal models in biomedical research [1]. Anesthetic protocols for safeguarding the animal’s welfare and refinement in medical investigations are essential [2,3]. To avoid stressing and restraining the pig during short procedures such as transport, imaging, and sampling, short-term anesthesia is often required in this species [4]. The challenge is to identify an anesthetic treatment that, without the use of sophisticated equipment, ensures an acceptable depth and length of anesthesia, a regular spontaneous breathing pattern, and a stable hemodynamic condition for the animal. Desirably, the anesthetic protocol should provide fast and reliable immobilization, adequate analgesia, and muscle relaxation without cardiovascular and respiratory depression [5]. Furthermore, prolongation of anesthesia must be possible without compromising the safety of the animal [6]. In terms of safety and efficacy in pigs, it has been reported that the combination of two or more drugs that target specific clinical effects is currently what constitutes the best standard for injectable intramuscular (IM) anesthesia [7]. Anesthesia with various combinations of α_2_-agonists, dissociative anesthetics, benzodiazepines, and opioids have been suggested as induction agents prior to general anesthesia or for use as short-term anesthesia [4,5,8,9]. In those previous studies, physiological parameters and time to unconsciousness and recovery have been presented without correlation to actual drug concentrations in the body. Drug concentration from a single injection of a certain drug and pharmacokinetic data have also been published [10,11,12]. However, when drug combinations are administered, drug interactions may occur and repeated anesthetics may further alter the physiology of the animal, which can impact the outcome of the results [6,13]. In a previous study, we have shown that the induction of anesthesia using an α_2_-agonist in combination with tiletamine-zolazepam and butorphanol administered IM produces a reliable and rapid anesthesia induction in pigs [14]. In that study, the anesthesia induction was followed by endotracheal intubation and general anesthesia; however, none of the factors such as the anesthesia length, the standard of recovery, or the drug plasma concentrations of the drugs were assessed. To the authors’ knowledge, the possible pharmacokinetic interactions between the drugs or the plasma concentrations and effects of these drug combinations used for induction of anesthesia have not yet been evaluated.

The aim of the study was to assess the adequacy and reliability of an anesthetic protocol and to evaluate physiological and clinical responses in relation to the drug plasma concentrations for growing pigs undergoing short-term anesthesia. A second aim was to see whether an intravenous (IV) dosage could be used to prolong the anesthetic time.

We hypothesized that a single IM and repeated IV injection of the drug combination in pigs will result in a rapid uptake and distribution of the drug mixture, as well as the maintenance of physiological parameters within acceptable ranges. 

## 2. Material and Methods

### 2.1. Animals and Housing

The experimental protocol was approved by the Ethics Committee for Animal Experimentation, Uppsala, Sweden (Approval No. C123/14, C2/16). The study design and reporting are in accordance with the ARRIVE guidelines. The study was performed at the Department of Clinical Sciences at the Swedish University of Agricultural Sciences, Sweden. The animals were handled according to the guidelines set out by the Swedish Board of Agriculture and the European Convention of Animal Care. A total of 12 pigs Yorkshire × Hampshire, of both sexes, were included in the study. Upon arrival from the university farm, they were 7–8 weeks old and weighed 25 ± 2 kg. The pigs were clinically examined before the start of the study and found to be healthy according to the American Society of Anesthesiologists (ASA 1). The animals were video recorded from the start of the acclimatization period until the end of the study. The animals underwent a 14-day acclimatization period at the Department of Clinical Sciences. They were housed in individual pens measuring 3 m^2^, where they could see and hear each other. Straw and wood shavings were provided as bedding. The ambient temperature inside the pens was 18 ± 2 °C, and a 10:14 h light–dark schedule was used. An infrared lamp was provided in a corner of each pen. The pigs were fed a commercial finisher diet (Solo 330 P SK, Lantmännen, Sweden) twice daily. The amount fed was according to body weight and the Swedish University of Agricultural Sciences (SLU) regimen for growing pigs. Water was provided ad libitum, and once a day, their general condition was examined. All of the pigs were included in a previously published study during the acclimatization period in which the animals were trained to accept touching and palpation of the ears in preparation for blood sampling from the auricular vein [14]. The pigs were randomly assigned into two groups using the R statistical software version 2.3.1 (GNU operation systems, Free Software Foundation, Boston, MA, USA): those who received a single IM injection (Group S) and those who received an additional injection IV (Group R), with *n* = 6 in each group.

### 2.2. Central Vein Catheterization 

Three days before the start of the study, an internal jugular catheter was placed under general anesthesia in all pigs to facilitate blood sampling. A light meal was given to the animals six hours before anesthesia, but water was provided ad libitum. Topical anesthesia (2.5 g) with lidocaine and prilocaine (EMLA 25 mg/g + 25 mg/g AstraZeneca, Södertälje, Sweden) was applied on the pigs’ ear flaps two hours before induction. Anesthesia was induced with sevoflurane (SevoFlo^®^ Orion Pharma, Danderyd, Sweden) in oxygen and air (FIO_2_ 0.5) that was delivered using a nonrebreathing system with a face mask. Initially, the fresh gas flow (FGF) was set at 300 mL/kg/min and the vaporizer at 8%. When the animal assumed a lateral recumbent position, FGF and the concentration of sevoflurane were decreased to 100 mL/kg/min and 4%, respectively. A polyurethane catheter (BD Careflow^TM^ 3Fr 200 mm, BD Medical, Franklin Lakes, NJ, USA) coated with MPC [14] was introduced via *V. auricularis* into *V. jugularis* using an aseptic Seldinger technique. The catheter was sutured onto the ear with monofil-coated polyamide (Supramid 2-0, B Braun Medical, Danderyd, Sweden) and covered with a bandage (Snøgg AS, Vennesla, Norway). Sevoflurane administration was discontinued at the end of the catheter placement, and the pigs were returned to their home pens to recover. The catheters were flushed with saline once daily until the start of the study. 

### 2.3. Anesthesia 

A light meal was given to the animals six hours before anesthesia, and water was provided ad libitum. The anesthesia was induced IM while the pigs were in their home pen using a butterfly needle (CHIRAFLEX Scalp vein set 21G × 3/4”, 0.8 × 20 mm Luer-Lock, CHIRANA T. Injecta, Stara Tura Slovakia). The pigs were given an IM injection in the brachiocephalic muscle on the side opposite the catheterized ear. One bottle of tiletamine and zolazepam (Zoletil 100^®^ vet. tiletamine 250 mg + zolazepam 250 mg, Virbac, Carros, France) in powder form was reconstituted with 5 mL of dexmedetomidine (Dexdomitor^®^vet. 0.5 mg/mL, Orion Pharma AB Animal Health, Danderyd, Sweden) and 1 mL of butorphanol (Dolorex^®^vet. 10 mg/mL, Intervet AB, Stockholm, Sweden). Thus, each milliliter contained 0.42 mg/mL dexmedetomidine, 83.3 mg/mL of tiletamine–zolazepam, and 1.67 mg/mL of butorphanol. The anesthesia dose for each pig was dexmedetomidine 0.025 mg/kg, tiletamine 2.5 mg/kg, zolazepam 2.5 mg/kg, and butorphanol 0.1 mg/kg (0.06 mL/kg of the anesthetic combination solution). The pigs were observed for any signs of discomfort during the injection, and their position and level of consciousness were monitored continuously. The time to unconsciousness, evidenced by lateral recumbency, head down, lack of reaction when manipulating or moving their body, and absence of the palpebral reflex, was noted. The trachea of the animals was not intubated and pigs breathed room air over the anesthetic period. Equipment for endotracheal intubation, laryngeal masks, self-inflating bag, and supplementary oxygen was prepared and available if cyanosis occurred or if the hemoglobin oxygen saturation (SpO_2_) decreased below 85%. After induction, the animals were positioned and supported on straw bedding to avoid discomfort or pressure injuries. During anesthesia, the animals’ bodies were checked for discomfort and manipulated or moved in the same position to another location in the pen every hour to mimic transport. The palpebral reflex, jaw tone, the occurrence of spontaneous movements, and response to stimuli, such as cannulation and blood sampling, were all used to determine the depth of anesthesia. The response to the nociceptive stimulus was elicited by mechanically clamping the coronary band of the medial or lateral dewclaw using forceps. The site of the stimulation was changed slightly each test to prevent sensitization to stimuli [15]. A cannula (BD Venflon™ 20 G × 32 mm, BD Medical, Franklin Lakes, NJ, USA) was inserted in the auricular vein on the noncatheterized ear 30 min after induction in both groups. Through this IV access, only Group R was given a further dose containing one-third of the initial dose, (0.02 mL/kg of the anesthetic combination solution) that was administered 60 min after induction. The time to first spontaneous movement, standing position, and the number of attempts to stand were observed and recorded. The quality of their recovery was assessed and recorded manually during the experiment, and at a later time reassessed by watching the video recordings. The assessment was made using a scoring system adapted and modified from an anesthesia scoring system used in antelopes that ranged from 1 (excellent) to 3 (poor) [16] (Table 1.) Any side effects observed during the procedure and the subsequent 72 h were recorded. 

### 2.4. Physiological Measurements and Blood Sampling 

Palpation and monitoring of the pulse (HR) and monitoring of respiratory rate (RR), SpO_2_ and rectal temperature (temp) (GE B40 Patient Monitor, GE Healthcare, Danderyd, Sweden) began once the animals were in lateral recumbency and continued throughout the anesthesia at the following intervals: 5, 10, 15, 30 and every 30 min until recovery.

Ethylenediaminetetraacetic acid (EDTA) tubes were used for the blood samples that were obtained via the central venous catheter before induction and repeated after at the following intervals: 5, 10, 15, 30, 60 min; 2, 3, 4, 5, 6, 7, 10 h; and finally, once daily, for up to two days after drug administration. Additional sampling was made when the pigs became unconscious, had a first spontaneous movement, and assumed a standing position. Plasma was separated by centrifugation in 5 min within 90 min and stored at −80 °C until analysis. 

### 2.5. Drug Analyses

The samples were prepared by mixing 50 μL of plasma with 100 μL of the standard internal solution used at the lab (50 ng/mL of midazolam and phenacetin in Acetonitrile), which were vortexed and centrifuged (Thermo SL16, 20 min, 4000 rpm). The samples were then transferred to a Waters 96-well plate and submitted for liquid chromatography–mass spectrometry analysis. Samples for making the standard curves were prepared into pig plasma by spiking the matrix into concentrations of 0.05–10,000 ng/mL of the tiletamine, 0.05–10,000 ng/mL of zolazepam, 0.005–1000 ng/mL of dexmedetomidine, and 0.005–1000 ng/mL butorphanol, and were otherwise treated as the samples.

Quality control (QC) samples were prepared into pig plasma by spiking the matrix into concentrations of 2, 20, 200, and 2000 ng/mL of tiletamine and zolazepam, but the high concentration QC samples were not used in the primary analysis since the calibration range extended only up to 1000 ng/mL. Samples were otherwise treated as samples. Quality control samples in pig plasma for dexmedetomidine and butorphanol were prepared in 0.2, 2, 20, and 200 ng/mL concentrations and were otherwise treated as samples. Samples with zolazepam concentrations higher than 1000 ng/mL were diluted and reanalyzed. Midazolam was used as an internal standard for the quantification of dexmedetomidine, tiletamine, and butorphanol. Zolazepam was quantified successfully without the use of an internal standard. 

### 2.6. Pharmacokinetic Analyses

For each animal and drug, the plasma drug concentrations vs. time were plotted. Different models and weighting factors were assessed by visual inspection of the curve fits and the residuals’ scatter plots, together with the accuracy of fit measures incorporated in the software, e.g., the Akaike information criterion. A noncompartmental model was used for all drugs. For Group S (IM), the maximal concentration of each drug in plasma (C_max_), time to reach C_max_ (t_max_), and terminal half-life (t_½_) were calculated with a noncompartmental model using the PK Solver add-in for Microsoft Office Excel. For Group R, after the IV bolus administration, the t_½_, the volume of distribution (Vd) and the clearance (cl) were obtained from the model.

### 2.7. Statistical Analysis 

Physiological and clinical data were compared among study times and groups using a one-way analysis of variance (ANOVA) for repeated measures. Data were presented as median, mean ± SD, or as a range. A *p*‒value of <0.05 was considered significant in all tests.

## 3. Results

The study was completed with a minor incident occurring in only one of 12 pigs. The drug concentration and pharmacokinetic analyses include data from a total of seven pigs (S = 3 and R = 4) since samples from five pigs were lost. Scoring and duration of anesthesia, as well as physiological and clinical results, include data from all 12 pigs (S = 6 and R = 6). 

### 3.1. Anesthesia

Induction of anesthesia was possible without the need to restrain any of the animals. The mean time from injection to lateral recumbency was 2.6 (±0.7) min, and from injection to unconsciousness, 10.0 (±3.7) min. During the anesthesia, excellent muscle relaxation was sustained until the first movement, and there were no reactions to cannulation, manipulation/movement of their body, and blood sampling in any of the animals. The withdrawal reflex was absent in 10 animals within 10 min after the induction, and it was present in two pigs for up to 60 min after the induction. The palpebral reflex was absent in all pigs 10 min after the induction. After the repeated injection IV, none of the pigs showed withdrawal reflexes when their dewclaw was mechanically clamped. The mean time from induction to the first spontaneous movement was 117 (±27) and 147 (±25) min, and from induction to standing position, 158 (±22) and 199 (±25) min, in Group S and R, respectively. In Group R, the time from the repeated injection to the first movement was 98 (±22) min, and to a standing position, 139 (±25) min (Table 2). The median recovery score was 1 (range 1–2) in both groups. 

### 3.2. Physiological Data

The mean values for HR and RR (*n* = 12) are shown in Figure 1. There were no differences in HR (*p* = 0.72) or RR (*p* = 0.073) over time in either group. During anesthesia, SpO_2_ was never below 86% (99–86%) in any of the pigs. One pig in Group R showed initial but transient apnea (about 10 s) immediately after the repeated injection IV. Rectal temperature decreased over time in both groups (range 40.1–37.3 °C), but no differences were seen between the groups.

### 3.3. Concentration and Pharmacokinetic Data

None of the catheters became obstructed or dislodged. No infection was observed at any of the catheter sites. None of the animals showed signs of local pain during palpation or blood sampling. Quality data for the analyses are presented in Table 3. Individual data on plasma concentration over time for each drug are shown in Figure 2. The main pharmacokinetic parameters are shown in Table 4, and observed clinical responses in relation to plasma concentrations are shown in Table 2. In all pigs, the average C_max_ for all drugs was reached 12 (range 10–13) min after administration of the IM induction drug combination. The highest concentrations in Group R were measured 5 min after the repeated injection IV (Figure 2). In Group R, the drug concentrations were 1.4–1.6 times higher 5 min after the repeated injection IV, compared to C_max_ after the IM induction (Figure 2). Within 15 min after the repeated injection IV, the mean plasma concentrations of the drugs in Group R were similar (tiletamine 336 ng/mL, dexmedetomidine 5.1 ng/mL, butorphanol 19.8 ng/mL, and zolazepam 1387 ng/mL) (Figure 2), compared to the C_max_ after IM administration. It was possible to detect the drugs up to 19 and 22 h for dexmedetomidine, 10 and 24 h for tiletamine, 29 and 31 h for zolazepam, and 23 and 31 h for butorphanol after administration of the induction dose in Group S and R, respectively. 

## 4. Discussion

The results of this study suggest that intramuscular administration of the drug combination dexmedetomidine, tiletamine, zolazepam, and butorphanol provided up to two hours of anesthesia with an acceptable level of analgesia, regular breathing pattern, and stable physiological parameters. Furthermore, at one hour of anesthesia, IV administration of one-third of the initial dose extended the anesthesia duration by another 30 min. 

### 4.1. Anesthesia

In the present study, all of the pigs were recumbent within 4 min and were unconscious within 15 min after the IM injection. This demonstrates a rapid uptake from the injection site resulting in a rapid onset of action. In addition, the observed C_max_ time of 10–13 min for all drugs after administration is in good agreement with the manually assessed time of unconsciousness. The time from drug administration to unconsciousness plays a critical role regarding the quality of the anesthesia [17]. Pigs are easily stressed and the time between injection and onset can cause ataxia during the induction phase [17,18]. Assessing the depth of anesthesia in pigs in the absence of surgical stimuli is challenging [19]. Clamping the dewclaw has been described in previous studies as an intense stimulus that persists until a high degree of central nervous system depression is reached, i.e., a supramaximal stimulus [20,21]. In those studies, response to dewclaw clamping was not consistent between the animals. Despite this, the authors stated that the method is a reliable indicator of anesthetic depth. In the present study, none of the 12 pigs reacted to cannulation, blood sampling, or being moved in the same position to another location in the pen, but there was variability in the withdrawal reflex, which was absent for up to 60 min in two of the animals. Hence, the levels of anesthesia and analgesia seem to be adequate for short procedures intended in this study. 

Muscle relaxation of the animals is crucial during transportation and diagnostic imaging procedures [22]. The animals in this study were characterized by complete muscle relaxation with no spontaneous movements until the recovery time. Alfa_2_ agonist sedatives are used in both veterinary medicine and biomedicine, and the drug has been described to provide central sedative effects, accompanied by muscle relaxation and analgesia when combined with tiletamine–zolazepam [3,23,24].

The pigs were not endotracheally intubated in the present study because the aim was to relate respiratory function to the plasma concentration of the drugs used and not to the effect of a supramaximal stimulus, i.e., intubation. Previous publications state that intubation is possible shortly after induction with a drug combination similar to the combination used in the present study [14,24,25]. In those reports, the induction was followed by maintenance with inhalation anesthesia and mechanical ventilation making intubation necessary. However, since intubation is a potent stimulus in the pig, the procedure may cause complications, e.g., changes in depth of anesthesia, reflexes, and breathing pattern. Additional anesthesia, such as fentanyl or propofol, may be needed in pigs induced with the combination of drugs used to suppress laryngeal reflexes and provide adequate relaxation for endotracheal intubation [26,27]. Under very deep sedation or anesthesia, complications such as hypoxemia, vomiting, and aspiration pneumonia may occur; therefore, endotracheal intubation is always recommended for pigs undergoing procedures under the described combination of drugs [28]. In case of complications, skilled staff and advanced equipment for prompt endotracheal intubation or placement of a laryngeal mask for the possibility to ventilate the pigs with a self-inflating bag and supplemental oxygen were prepared.

Moreover, 15 min after the repeated injection IV, the plasma concentrations of all drugs were similar to the C_max_ after the initial IM administration. One-third of the initial dose prolonged the anesthesia by approximately 26% when given IV to the animals. The plasma concentration ranges of each drug from the last blood sample that was taken while the pigs were still unconscious are overlapping for zolazepam and butorphanol, but not for dexmedetomidine and tiletamine (Table 4). There is no doubt that these drug concentrations predict such events due to dexmedetomidine’s sedative effect and tiletamine’s dissociative anesthetic effect. Moreover, since the four drugs were given in combination, the effects of concentrations could not be calculated in this study. Additionally, the t_max_ results for all drugs in this combination were rather similar, and as a result, they will decline in a similar matter. In polypharmacy, the drugs may have an impact on each other’s pharmacokinetic profiles, and a shorter t_max_ for zolazepam and tiletamine was observed in this study, compared to other studies in pigs when only zolazepam and tiletamine were given [10,11,29]. In this study, it is not possible to link each drug’s pharmacokinetics to pharmacodynamics, but rather, the combined effect of all drugs together for use in robust short-term anesthesia has been discovered and described. The sole adverse event observed was a short episode of apnea (about 10 s) that occurred immediately after giving the repeated injection IV containing one-third of the initial dose. It was easily managed, and the pig started to breathe spontaneously after the application of light pressure on its chest. Therefore, it was decided to include the pig in the analyses.

Over the course of the anesthesia and even after the repeated injection, the physiological measures remained within normal limits. This was unexpected because α_2_-agonists commonly produce bradycardia [30]. However, tiletamine that was included in the induction combination increases HR by stimulating the sympathetic nervous system [31].

Since the observer was aware of the animals’ treatments, the scoring system was based on what was written in the notes, as well as the number of attempts to stand and the degree of ataxia seen in the video recordings. The median recovery score was excellent (score 1), with the animals making one or two attempts to stand with very mild signs of excitement. Two animals, one in each group, had a good recovery (score 2), but none of the animals was scored as poor recovery (score 3) (Table 1). Overall, the drug combination was successful in producing a good to excellent recovery. 

Hypothermia is a common anesthesia complication, and it occurs due to altered thermoregulatory control as well as evaporative and conductive heat loss [32]. The decrease in temperature occurring in both groups could have probably been avoided with the use of heating pads and blankets. However, the body temperature did not fall below 37.3 °C in any animal, which is within the normal range for pigs undergoing anesthesia [33].

The initial IM dose was based on previous studies conducted on pigs at our research laboratory. The repeated dose used was based on a procedure in a similar population of animals where catheters were placed via subcutaneous tunneling. In that procedure, an IV injection after one hour of anesthesia containing one-half the initial dose of the combination resulted in apnea (>10 s) in approximately 50% of the pigs. When one-third of the initial dose was used, sufficient spontaneous respiration was maintained, and reflexes from catheterization stimuli were absent.

### 4.2. Pharmacokinetics

There were no t_½_ differences in regard to route of administration for any of the drugs. After an extravascular drug administration, such as IM in this case, the t_½_ can be more prolonged than after an IV administration. Thus, the route of administration for a drug can be one reason for the prolongation. This (flip-flop phenomenon) occurs when the rate of absorption is the rate-limiting step in the sequential processes of drug absorption and elimination. The drug cannot be eliminated before it has been absorbed, and the t_½_ of the absorption depends on the disappearance of the drug from the site of administration, which, in turn, depends on the physiological absorption process and how much of the drug is bioavailable.

Since the process of absorption does not seem to be a limiting factor for this drug combination, the t_½_ expresses the overall rate of the actual drug elimination process during the terminal phase. This overall rate of elimination depends on drug clearance and on the extent of drug distribution.

We were unable to compare pharmacokinetic studies for this drug combination. The pharmacokinetic parameters for dexmedetomidine (or medetomidine and detomidine) and butorphanol in pigs seem to be unpublished, and only a few reports for tiletamine and zolazepam are available. There are similarities between the results for butorphanol in a goat study [34], in which the Vd was 1.27 L/kg, cl 0.0096 L/kg/min and t_½_ 1.87 h with IV administration, and for the pigs in the present study1.6 L/kg, 0.012 L/kg/min and 1.68 h. After IM administration, the time to C_max_ was comparable (t_max_ 16 min), but the t_½_ was longer (2.75 h in the goat), suggesting a flip-flop phenomenon in the goat that was not observed in the pigs in the present study. The goats received only butorphanol and became hyperactive within the first 5 min after administration, which was not the case for the pigs since they received butorphanol in a combination of drugs meant for short-term anesthesia.

Polypharmacy, as well as other variables including species and age, may affect a drug’s pharmacokinetics, making comparisons difficult. In another study [35], the IV pharmacokinetics of butorphanol, detomidine, and a combination of both administered to horses revealed that butorphanol given alone showed about a twofold larger clearance but similar t_½_, when compared with the combination. When detomidine was administered alone and compared with the combination, there was a similar clearance but a slightly shorter t_½_. The t_½_ for butorphanol in the horse (mean 5.2 h as a single drug and 5.4 h in the combination) is considerably longer than in the pig. There is a similar clearance (0.01 L/kg/min) when it is given alone but noticeably lower (0.006 L/kg/min), when compared with the combination of both.

A clearance of about 0.01 L/kg/min for butorphanol was also found in a study involving horses [36]. It has been suggested that the t_½_ is influenced by a physiological compartment in the horse that can be saturated and that a lower dose of butorphanol is sufficient to be efficacious when used in combination with the α_2_-agonist detomidine [35]. In pigs, the t_½_ for dexmedetomidine was in our study almost two hours, which is longer than the t_½_ (less than one hours) in horses, dogs, and cats [36,37,38,39,40] In the present study, only the pharmacokinetics of the combination with the α_2_-agonist dexmedetomidine was explored. It is yet to be investigated if a similar influence by the combination also exists in pigs.

An equal-dose combination of tiletamine and zolazepam is often used for the induction of short-term anesthesia in various species of animals, but information regarding its pharmacokinetics and metabolism is scarce. The anesthetic effects of the drugs may differ from species to species depending on the elimination and metabolic clearance. The plasma concentrations in the present study are consistent with the data obtained in a previous study [11] that showed higher concentrations of tiletamine–zolazepam than in pig plasma from 16 Yorkshire-crossbred pigs given a single dose of 3 mg/kg tiletamine and zolazepam IM. The zolazepam had a lower t_½_ (zolazepam 2.76 h versus tiletamine 1.97 h) and clearance, compared to the tiletamine, which demonstrates the major pharmacokinetic and metabolic differences between the two drugs. However, the findings of the current study did not appear to have this difference in t_½_ (zolazepam 1.2 h versus tiletamine 1.5 h) after administration IM. Our findings may be a result of the combination of drugs that were similar to those found in a study of pregnant pigs [41], where the C_max_ for both tiletamine and zolazepam was 50–60 min (compared to 10–12 min in the present study), and where the elimination of zolazepam was slow and tiletamine rapid. The study also showed that zolazepam, but not tiletamine, was detected in the uterus and umbilical cord and thereby probably having an effect on the fetus; this finding must be taken into consideration when used in pregnant sows.

Three metabolites of zolazepam and one metabolite of tiletamine in plasma, urine, and microsomal incubations were analyzed in a study [11]. Unfortunately, the muscle-relaxant, antianxiety, and sedating properties of the metabolites of zolazepam were not reported. The metabolism and plasma clearance of zolazepam was reported to be slower than tiletamine, thus leading to prolonged sedative and muscle-relaxant effects. For the welfare of the pig, and as long as good analgesia/anesthesia during the intervention is provided, this must be considered to be a better scenario than vice versa. Since the effect of tiletamine alone as a dissociative anesthetic does not provide muscle relaxation, it can cause a cataleptic state. The t_½_ of tiletamine has been reported to be shorter than that of zolazepam in other animal species. In a polar bear study, an average t_½_ of 1.8 h and 1.2 h, for tiletamine and zolazepam, respectively, was reported [42].

It has been rereported in a review [29] that tiletamine has a t_½_of 2–4 h in cats, 1.2 h in dogs, 1–1.5 h in monkeys, and 30–40 min in rats. For zolazepam, the reported t_½_ results are 4.5 h in cats, 4–5 h in dogs, 1 h in monkeys, and 3 h in rats. Comparable pharmacokinetic properties of fixed-dose combination drugs may be desirable in order to permit similar dosing intervals. For anesthetic/muscle-relaxant combinations, the pharmacodynamics/anesthetic effects such as dissociative anesthesia and muscle relaxation should also be considered. According to the same review [29], the explanation for using a 1:1 dose ratio was based on pharmacodynamic considerations from studies performed on cats, dogs, and monkeys, despite differences in the t_½_ of tiletamine and zolazepam in these species.

## 5. Limitations

The number of animals included was based on a previous sample size calculation. However, the sample size was decreased by the lost plasma samples. In addition, relatively high individual variability limited the power to detect significant differences that might have been found if all samples had been included. Before the start of the present study, the pigs had been involved in a training program for two weeks. The pig training, which lowered patient stress levels, may have contributed to the short period between injection and unconsciousness, as well as the excellent recovery rates. Different findings can be obtained in laboratories that do not have this stress-reduction program.

It should be emphasized that arterial blood gas analysis remains ideal for the assessment of oxygenation and to address the presence of hypoxemia [43,44]. In the present study, arterial blood gas analysis was not performed. Arterial catheterization usually requires surgical exposure of the deeply located vessels that would likely produce a potent stimulus in the pigs. Due to the small sample of animals used and the use of pulse oximetry instead of blood gas analysis, further studies are warranted to investigate the presence of hypoxemia.

## 6. Conclusions

In conclusion, the combination of dexmedetomidine, tiletamine, zolazepam, and butorphanol, given intramuscularly, provides a rapid induction of good quality, followed by up to two hours of anesthesia with acceptable tolerability to nociceptive stimulus in growing pigs. The plasma concentration profile of the drugs was in line with the duration of the effect, showing a rapid uptake and distribution of the drugs. Since the technique used ensures a satisfactory breathing pattern and physiological stability, this procedure is suitable for the care and health of pigs. In addition, at one hour of anesthesia, intravenous injection of one-third of the original dose extended the anesthesia duration by another 30 min.

## Figures and Tables

**Figure 1 animals-11-01482-f001:**
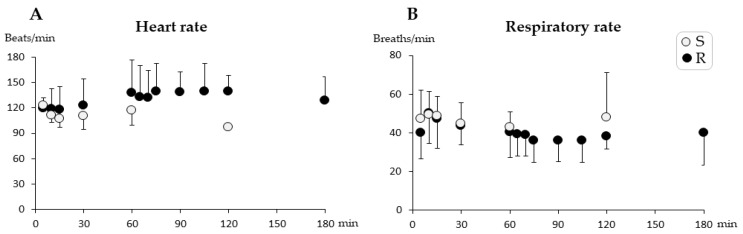
Heart rate (**A**) and respiratory rate (**B**) (mean ± SD) during short-term anesthesia recorded at 5, 10, 15, 30, and every 30 min from injection until recovery. Pigs from Group Single (S) (*n* = 6) received a single dose intramuscularly of dexmedetomidine 0.025 mg/kg, tiletamine 2.5 mg/kg, zolazepam 2.5 mg/kg, and butorphanol 0.1 mg/kg at time zero. Pigs from Group R (*n* = 6) also received one-third of the initial dose intravenously at 60 min. No significant differences were found between groups or over time.

**Figure 2 animals-11-01482-f002:**
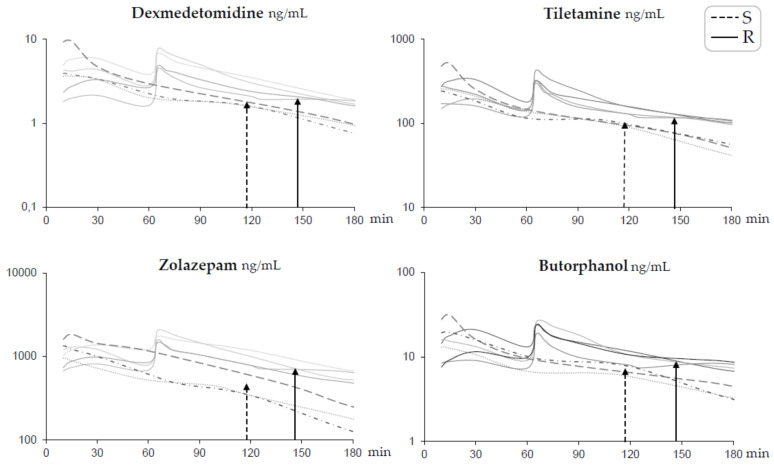
Profiles of log plasma concentrations for each drug over time for short-term anesthesia induced by intramuscular administration of dexmedetomidine 0.025 mg/kg, tiletamine 2.5 mg/kg, zolazepam 2.5 mg/kg, and butorphanol 0.1 mg/kg in pigs (*n* = 7). Group S (dotted lines *n* = 3) received a single dose at time zero. Group R (solid lines *n* = 4) also received one-third of the initial dose intravenously at 60 min. Time to first spontaneous movement is indicated with a dotted arrow for Group S and a solid arrow for Group R.

**Table 1 animals-11-01482-t001:** Recovery Scoring System.

Recovery Score	Description
1 Excellent	Animal transitions from lateral to sternal recumbency with minimal ataxic movements. Stands in one or two attempts. Walks with only slight ataxia.
2 Good	Animal in transition from lateral to sternal recumbency displays moderate ataxic movements and may take one or two attempts. Some imbalance in sternal recumbency and requires more than two attempts to stand. Walks with moderate ataxia.
3 Poor	Animal makes frequent attempts with severe ataxic movements to transition from lateral to sternal recumbency before being successful. Severe imbalance in sternal recumbency. Makes numerous attempts to stand but falls before being successful and displays marked ataxia when walking.

Scoring system (1–3) used to categorize the quality of recovery in pigs.

**Table 2 animals-11-01482-t002:** Observed clinical responses in relation to plasma concentrations.

			Concentration Range (ng/mL) Group S and R			
Noted Observations	Time (min) from Injection (IM)		Dexmedetomidine	Tiletamine	Zolazepam	Butorphanol
	S	R				
Lateral recumbency	2–4		<1.59–3.56	<126–268	<106–789	<5.22–15.60
Unconsciousness	5–15		1.99–6.00	169–337	737–1330	8.86–19.20
Last sample before response to noxious stimulus	60–120	120	2.01–3.48	115–158	520–1180	6.74–11.60
Palpebral reflexes	60–70	126–180	1.69–2.38	101–129	525–782	6.80–10.50
Response to noxious stimulus	70–140	126–180	1.69–2.38	101–129	525–782	6.80–10.50
First movement	70–140	126–180	1.69–1.97	106–123	444–713	6.47–9.42
Standing	120–180	175–240	1.14–2.01	73.7–109	250–678	4.74–8.70

Time range in min and the corresponding plasma concentration range for each drug and noted observation during the short-term anesthesia induced by dexmedetomidine 0.025 mg/kg, tiletamine 2.5 mg/kg, zolazepam 2.5 mg/kg, and butorphanol 0.1 mg/kg after intramuscular (IM) administration in growing pigs single (S) (*n* = 6) and repeated (R) (*n* = 6). All pigs received a single dose at time zero, and group R also received one-third of the initial dose intravenously at 60 min. All blood samples were taken in relation to the noted observation except for lateral recumbency, which was taken at 5 min after induction. Plasma concentrations were available for seven pigs.

**Table 3 animals-11-01482-t003:** Drug analysis data.

Compound	Dexmedetomidine	Tiletamine	Zolazepam		Butorphanol
Detection limit (ng/mL)	<0.02	0.2	0.05	0.05	0.02
Quantitation limit (ng/mL)	0.02	0.5	0.1	0.1	0.05
Range (ng/mL)	0.02–200	0.5–1000	0.1–1000	0.1–500	0.05–500
R^2^	>0.999	>0.998	>0.997	>0.996	>0.999

Detection limit (LoD), quantitation limit (LoQ), range of standard curve, and coefficient of determination R-squared value (R^2^) of each drug.

**Table 4 animals-11-01482-t004:** Pharmacokinetic parameters.

	IM Group S (*n* = 3)			IV Group R (*n* = 4)		
	t_½_ (min)	t_max_ (min)	C_max_ (ng/L)	t_½_ (min)	cI (L/kg/min)	Vd (L/kg)
Dexmedetomidine	125 ± 37	13 ± 3	5.6 ± 3.2	117 ± 24	0.012 ± 0.004	1.7 ± 0.5
Tiletamine	90 ± 12	10 ± 5	342 ± 152	80 ± 13	0.050 ± 0.010	5.8 ± 0.9
Zolazepam	72 ± 6	12 ± 3	1372 ± 438	76 ± 13	0.009 ± 0.002	1.0 ± 0.2
Butorphanol	97 ± 11	13 ± 3	21 ± 9	101 ± 16	0.012 ± 0.002	1.6 ± 0.4

## Data Availability

The data presented in this study are available on request from the corresponding author. The data are not publicly available due to privacy.

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
