# Peer review of "Physiological and Clinical Responses in Pigs in Relation to Plasma Concentrations during Anesthesia with Dexmedetomidine, Tiletamine, Zolazepam, and Butorphanol"

_animals, 2021, doi:10.3390/ani11061482_

Round 1

Reviewer 1 Report

Dear Author,

Thank you for submitting the manuscript Physiological and clinical responses in pigs in relation to plasma concentrations during anaesthesia with dexmedetomidine, tiletamine, zolazepam and butorphanol to Animal Journal.

Please accept my suggestions listed below:

The author needs to state clearly what is the hypothesis of the study. It is important that study power is discussed when interpreting findings from the study and discussing its shortcomings. Did the authors perform sample power calculation when designing the study?

Is the study conform with the ARRIVE guidelines?

I do have some concerns regards the fact that some pigs experienced hypoxemia.Low oxygen saturation is associated in non-intubated patients with reduced oxygen partial pressure in inspired air. There is no clear cut-off point for abnormal oxygen saturation, but SpO2 ≤ 95% is used in most human and animal studies. Observation of mucous membrane colour is not a sensitive indicator of hypoxemia as cyanosis will likely not occur until hypoxemia is profound (Kelman, 1966). Hypoventilation can cause hypoxemia, the time of recumbency of these patients is quite long and some degree of atelectasis would definitely occur. I see that the respiratory rate did not change in the study, but shallow breaths, at the same rate may not be efficient to ensure enough oxygen transport to the alveoli. Hypoventilation can be estimated by observing respiratory rate and depth (which is very subjective). Hypoventilation can cause hypercarbia, with subsequent respiratory acidosis, and hypoxemia, but capnometry/graphy was not monitored in the study. In this regard, SPO2 is not clinically suitable as a surrogate for PaO2 (Farrell et al. 2019). There are papers looking at SpO2 /FiO2 ratio for the evaluation of anaesthetized patients on supplemental oxygen. However arterial blood gas analysis remains ideal for the assessment of oxygenation (Farrell et al.2019) and the most reliable method to address the presence of hypoxaemia. This should state in the discussion suggesting further studies to prove it.

On-Line 317 over anaesthesia discussion the author wrote that: “We observed neither excessive salivation nor vomiting and the SpO2 was satisfactorily preserved in the animals.” The number of animals used is very low to state this. Bigger sample size should have been used to evaluate this particular condition. Again, it should be stated in the discussion.

Please see the list below for rewording and error suggestions.

Summary:

Line 10: The study's aim is to identify an anaesthetic procedure that, without the use of sophisticated equipment, ensures an acceptable depth and length of anaesthesia, a regular spontaneous breathing pattern, and a stable hemodynamic condition for the animal.

Line 16: The main results suggest that intramuscular administration of the drug combination provides up to two hours of anaesthesia with stable physiological parameters and an acceptable level of analgesia

Line 18: intravenous administration of one-third of the original dosage prolonged the anaesthesia for another 30 minutes.

Line 20: Since the patients were able to breathe spontaneously, any of them was intubated.

Line 21: The study also provides new information about each drug's plasma concentrations and the impact of the drug combination in pigs. This technique can be used to perform non-surgical operations or transports when short-term anaesthesia is required.

Abstract

Line 26: The aim of the study was to assess the adequacy and reliability of an anaesthetic protocol as well as to evaluate physiological and clinical responses, in relation to the drug plasma concentrations, for growing pigs undergoing short-term anaesthesia.

Line 28: A second aim was to see whether an intravenous dosage could be used to prolong anaesthesia.

Line 31: After one hour, six of the pigs received a repeat intravenous injection of one-third of the initial dose.

Line 34 Each drug’s absorption and time to maximal concentration were rapid.

The rest of the paragraph was rearranged:

All pigs were able to maintain spontaneous respiration. There were no differences in the half times in regard to the route of administration. The results suggest that intramuscular administration of the four-drug combination provides up to two hours of anaesthesia with stable physiological parameters and an acceptable level of analgesia while maintaining spontaneous respiration. Repeated intravenous injection may also be used to extend the time of anaesthesia by 30 minutes.

Introduction

Line 50: The challenge is to identify an anaesthetic treatment that, without the use of sophisticated equipment, ensures an acceptable depth and length of anaesthesia, a regular spontaneous breathing pattern, and a stable hemodynamic condition for the animal.

Line 74: have not yet been evaluated

Line 75: The aim of the study was to assess the adequacy and reliability of an anaesthetic protocol as well as to evaluate physiological and clinical responses, in relation to the drug plasma concentrations, for growing pigs undergoing short-term anaesthesia. A second aim was to see whether an intravenous dosage could be used to prolong the anaesthetic time.

Material and Methods

Line 86: 12 pigs Yorkshire x Hampshire, 7–8 weeks old, weighing 25 ± 2 kg were included in the study.

Line 92 They were housed in 3 m2 individual pens where they could see and hear each other.

Line 93: The ambient temperature inside the pens was 18 ± 2 °C, maintained by the use of an infrared lamp and a 10:14 h light-dark schedule was used.

Line 101: The pigs were randomly assigned into two groups using the R statistical software version 2.3.1: those who received a single IM injection (Group S) and those who received multiple IV injections (Group R), with n = 6 in each group.

Line 108: and

Line 129: zolazepam

Line 134: The calculation of tiletamine and/or zolazepam per mg/ml in a 6 ml solution is 8.3mg/ml for each drug, therefore the dose mg/kg in the solution is 0.5 mg/kg, please amend the error.

Line 136: zolazepam

Line 136: the calculation of the tiletamine or zolazepam is not correct:

Dilution of 50mg tiletamine + 50mg zolazepam in 6 ml (5ml dexmedetomidine + 1 ml butorphanol) = 16.6 mg/ml tiletamine/zolazepam or 8.3mg/ml tiletamine and 8.3mg/ml zolazepam. Therefore in 0.06 ml of anaesthetic combination solution there is 1mg of tiletamine/zolazepam or 0.5mg tiletamine and 0.5mg zolazepam. The mistake is also seen in the Table 4 caption and Figure 1 caption.

This need to be clarified and amended.

Line 150: response to the nociceptive stimulus was elicited by mechanically

Line 155: was given a further dose

Line 156: the dose of tiletamine and zolazepam needs to be amended since 1/3 of 0.5mg/ml is 0.166 mg/ml.

Line 159: The quality of their recovery was assessed and recorded manually during the experiment and re-evaluated through the recordings.

Line 169: The author needs to specify when the physiological parameters were recorded (q 5, q 10 minutes etc.) The same needs to be specified in the caption of Figure 1.

Results

Table 4

Line 241:The tiletamine/zolazepam calculation error is present again in the caption

Over Line 134, 136, 156 the calculation of the tiletamine/zolazepam is not correct.

The error is present in the Table 4 caption and Figure 1 caption. It is important to understand if the author administered the written doses or it is only a calculation error.

It is confusing to understand where the values of Group R are reported in Table 4.

Is it possible to use a separate Table to describe the time of the note observations (first movement and standing position) for Group R?

Line 249: how long was the apnoea?

Discussion

Line 280: with an acceptable level of analgesia, regular breathing pattern and stable physiological parameters.

Line 298: the author mentioned changing of position of the pigs, it should be mentioned also in materials and methods. Why was the body position changed? This could further affect the atelectasis

Line 314: Needs to highlight: complications such as hypoxaemia, vomiting, and aspiration pneumonia may occur under very deep sedation; other methods of avoiding hypoxaemia, at the very least, would have included the use of a laryngeal mask and oxygen supplementation.

Line 330: Please change ‘since they were given’ with since the four drugs were given in combination…

Line 339: Can the author please state how long the apnoea lasts?

Line 342: The physiological measurements remained within normal limits after the anaesthesia was prolonged and even after the repeated dosage.

Line 347 Please remove different

Line 392: Polypharmacy, as well as factors such as species and age, may influence a drug's pharmacokinetics, rendering comparisons challenging.

Limitations:

Line 452: The pig training, which lowered patient stress levels, may have contributed to the short period between injection and unconsciousness, as well as the excellent recovery rates. Some labs that do not have this stress-reduction preparation have different results.

Conclusion

Line 459: Please change antinociception with: acceptable tolerability to mechanical pain

Line 463: In addition, at one hour of anaesthesia, intravenous injection of one-third of the original dose extended the anaesthesia duration by another 30 minutes.

References:

Reference 15 is not pertinent to the manuscript

Reference 31 is not suitable for that sentence.

Reviewer 2 Report

The manuscript “Physiological and clinical responses in pigs in relation to 2 plasma concentrations during anaesthesia with dexme-3 detomidine, tiletamine, zolazepam and butorphanol” gives an overview of a simple protocol for short term anaesthesia in pigs, and investigated the physiological response in relation to plasma concentrations. The aim was to investigate the aesthetic effect of a single injection og the combined drug and if the anesthetic period could be prolonged with a follow-up dose of one third of the original dose, and to measure plasma concentrations of the individual drugs after combined dosing.

The study is well described, and the methods reflect a well-designed study. One question is, why the pigs were not intubated. Generally, intubation during anaesthesia is performed as a safeguard in case anesthetic complications occur during anaesthesia without assisted ventilation, and manual ventilation, e.g. with a balloon, is necessary. This possibility is not present when the animal is not intubated. In the Discussion, the manuscript cites reference 27, but to the reviewer opinion this quote is taken out of its context. Indeed, the reference states that there are grounds to avoid intubation, but it is also underlined that suitable combinations of anesthetic drugs should be used if intubation is needed. This reference illustrates the importance of intubation well, as two pigs experienced apnoea and needed positive pressure ventilation – which only could be performed because the animals were intubated. The manuscript returns to the topic of intubation in the Discussion, and reference 27 is cited again. The reviewer suggests balancing the discussion, by indicating situations where intubation can be safely omitted.

The following questions are asked out of curiosity, but have no influence on the review:

Was there a particular reason that a score system for recovery og antelopes was used? To the reviewer’s knowledge, there are only few anatomical and physiological similarities between antelopes and pigs.

Why was heart rate palpated and not taken from the reading of the pulsoximeter? Surely a pulsoximeter gives a more precise reading?

Required minor edits:

Please note that detomidine and tiletamine are spelled with an e.

Please note that there is a typo in Table 3, IM grop S instead of IM group S.

Tables 2 and 3 should be presented in the Results and not under Materials and Methods. Reference is made to Table 2 under Results, but no mentioning is made of Table 3. Please describe Table 3 briefly in the results section 3.3.

It is unclear from the table text if the results in Table 4 originates from 7 or 12 pigs. If data is based on 7 animals, please delete (n=12) and change to “Plasma concentrations were available for 7 out of 12 pigs”.

Figure 2 is presented in the text before Figure 1. Generally, figures are to be presented in numerical order.

In Figure 2, no legend is given for the solid grey lines. Please include this in the legend, and, also, indicate in the Figure text what the arrows indicate.

One pig in Group R showed initial, transient apnoea immediately after the repeated injection IV. Please state if this pig was included or excluded from the analyses.

Round 2

Reviewer 1 Report

Dear Author,

The study appears to have improved significantly in terms of content; however, I would recommend some slight language improvements to make the manuscript more reader-friendly. Thank you

Keywords: Please remove blood sampling, pharmacodynamics, refine

Line 30: was given

Line 67: produces a reliable and quick anaesthesia induction.

Line 68: In that study, the anaesthesia induction was followed by endotracheal intubation and general anaesthesia, however neither the anaesthesia length, nor the standard of recovery, nor the drug plasma concentrations were assessed.

Line 78: We hypothesised that a single and repeated IV injection of the drug combination in pigs will result in rapid uptake and distribution of the drug mixture, as well as the maintenance of physiological parameters within acceptable ranges, without the need for endotracheal intubation.

Line 87: The animals were handled according to the guidelines set out by the Swedish Board of Agriculture and the European Convention on Animal Care.

Line 114: from an anaesthetic breathing system (Flow-i Anesthesia Machine, Getinge, Swe- den) via an anaesthetic face mask

Please remove the above sentence and leaves only:

Anaesthesia was induced with sevoflurane (SevoFlo® Orion Pharma, Danderyd, Sweden) in oxygen and air (FIO2 0.5) delivered by a non-rebreathing system by face mask

Line 122: General anaesthesia was stopped at the end of the catheter insertion and the pigs were returned to their home pens to recover.

Line 130: The pigs were given an IM injection in the brachiocephalic muscle on the side opposite the catheterized ear.

Line 135: please remove the sentence “ in place of the water that is recommended by the manufacturer”

Line 143: The animals were not intubated and breathed room air over the anaesthetic period.

Line 148: During anaesthesia the animals' bodies were checked for discomfort and manipulated or shifted every hour to mimic transport.

Line 150: The palpebral reflex, jaw tone, occurrence of involuntary movements, and response to stimuli, such as cannulation and blood sampling, were all used to determine the depth of anaesthesia.

Line 158: Please can the author just leave the sentence: Over this intravenous access was given a further dose containing one-third of the initial dose, (0.02 mL/kg of the anesthetic combination solution) that was administered IV 60 minutes after induction. It makes the sentence more fluent. Thanks

Line 162: Please remove : and re-evaluated through the recordings. It does not make sense otherwise

Line 252: analysis

Line 288: Furthermore, at one hour of anaesthesia, intravenous administration of one-third of the initial dose extended the anaesthesia duration by another 30 minutes.

Line 304: In the present study, none of the twelve pigs reacted to cannulation, blood sampling, or being moved in the pen, but there was variability in the withdrawal reflex, which was absent for up to 60 minutes in two of the animals.

Line 310: The animals in this study were characterised by complete muscle relaxation with no spontaneous movements until the recovery time.

Line 315; In the present study, pigs were not intubated.

The pigs were not endotracheally intubated in the present study because the aim was

to relate respiratory function to the plasma concentration of the drugs used and not to the effect of a supramaximal stimulus, i.e., intubation. Previous publications state that intubation is possible shortly after induction with a drug combination similar to the combination used in the present study [14,24,25]. In those reports, the induction was followed by maintenance with inhalation anesthesia and mechanical ventilation making intubation necessary. However, since intubation is a potent stimulus in the pig, the procedure may cause complications e.g., changes in depth of anesthesia, reflexes and breathing pattern. Additional anaesthesia, such as fentanyl or propofol, may be needed in pigs induced with the combination of drugs used to suppress laryngeal reflexes and provide adequate relaxation for endotracheal intubation [26,27]. Under very deep sedation or anaesthesia, complications such as hypoxemia, vomiting, and aspiration pneumonia may occur, so endotracheal intubation is always recommended for pigs undergoing procedures under the described combination of drugs

In case of complications, skilled staff and equipment for prompt endotracheal intubation or placement of a laryngeal mask and possibility to ventilate the pigs with a self-inflating bag and supplemental oxygen was prepared.

Line 338: It's not doubtful that these drug concentrations predict such events due to dexmedetomidine's sedative effect and tiletamine's dissociative anaesthetic effect.

Line 341:…as a result, they will decline in a similar manner.

Line 351: analysis

Line 352: Over the course of the anaesthesia and even after the repeated injection, the physiological measures remained within normal limits.

Line 356: Since the observer was aware of the animals' treatments, the rating system was based on what was written in the notes, as well as the number of attempts to stand and the degree of ataxia seen in the video recordings.

Line 401: Polypharmacy, as well as other variables including species and age, may affect a drug's pharmacokinetics, making comparison difficult.

Line 464: Different findings can be obtained in laboratories that do not have this stress-reduction planning.

Line 474: Since the technique used ensures a satisfactory breathing pattern and physiological stability, this procedure is suitable for the care and health of pigs.
